# Internet Access and Nutritional Intake: Evidence from Rural China

**DOI:** 10.3390/nu13062015

**Published:** 2021-06-11

**Authors:** Ping Xue, Xinru Han, Ehsan Elahi, Yinyu Zhao, Xiudong Wang

**Affiliations:** 1Institute of Agricultural Economics and Development, Chinese Academy of Agricultural Sciences, Beijing 100081, China; xueping1017@hotmail.com (P.X.); wangxiudong@caas.cn (X.W.); 2School of Economics, Shandong University of Technology, Zibo 255049, China; ehsanelahi@cau.edu.cn; 3College of Economics and Management, Northwest A&F University, Yangling 712100, China; zhaoyy1127@163.com

**Keywords:** internet access, nutritional intake, rural China, propensity score matching

## Abstract

Over the past 4 decades, China has experienced a nutritional transition and has developed the largest population of internet users. In this study, we evaluated the impacts of internet access on the nutritional intake in Chinese rural residents. An IV-Probit-based propensity score matching method was used to determine the impact of internet access on nutritional intake. The data were collected from 10,042 rural households in six Chinese provinces. The results reveal that rural residents with internet access have significantly higher energy, protein, and fat intake than those without. Chinese rural residents with internet access consumed 1.35% (28.62 kcal), 5.02% (2.61 g), and 4.33% (3.30 g) more energy, protein, and fat, respectively. There was heterogeneity in regard to the intake of energy, protein, and fat among those in different income groups. Moreover, non-staple food consumption is the main channel through which internet access affects nutritional intake. The results demonstrate that the local population uses the internet to improve their nutritional status. Further studies are required to investigate the impact of internet use on food consumed away from home and micronutrient intake.

## 1. Introduction

Ending malnutrition is one of the main targets of the Sustainable Development Goals (SDGs) [1]. China has experienced a nutritional transition over the past 4 decades [2,3,4,5,6]. In China, both urban and rural residents are switching from low-fat, traditional food, mainly based on cereals and vegetables, with few animal products, to a Western-style diet that is high in saturated fat and sugar and low in fiber [7,8,9]. 

Evidence shows that nutritional intake is determined by income [8,10,11], agricultural/food programs [12,13,14], agricultural commercialization [15], microcredit [16], farm production [17,18], nutrition labels, and communication networks [19]. Moreover, studies have found that the internet is a new factor affecting the well-being of households, especially rural households, in both developed and developing countries/regions [20,21,22,23,24,25]. Specifically, the internet may positively affect food consumption [26,27,28,29,30].

In the last 2 decades, with the rapid development and widespread application of the internet, China has developed the largest population of internet users in the world. According to the China Internet Network Information Center (CNNIC), by the end of 2018, the number of Chinese Internet users (netizens) reached 829 million, and 222 million of them were rural residents [31]. It is surprising that the rural population of China that is connected to the internet is equal to the combined populations of France, Germany, the UK, and Australia. The CNNIC pointed out that the internet has already affected the lives of rural Chinese residents [31].

This article contributes to the existing literature in three ways. First, to the best of our knowledge, with the exception of the work of Parlasca et al. [32], the previous literature seldom investigates the relationship between internet access and nutritional intake. Using household panel data, Parlasca et al. [32] proved that mobile phone adoption and use were positively and significantly associated with dietary diversity. Although mobile phones were the main devices by which farmers accessed the internet, they could not be simply treated as a proxy variable for the internet. Moreover, the National Bureau of Statistics of China (NBSC) showed that only parts of mobile phones connect to the internet through cellular data or broadband networks (WiFi) in rural areas [33]. Thus, we explored the effects of the internet rather than the effects of mobile phones. Second, contrary to previous studies [22,23,29], we mainly studied the relationship between internet access and nutritional intake in order to assess the effects that internet access has on well-being more intuitively. As a result of the nutritional transition process and nutrition-related health problems in China [34,35,36], it is essential to investigate the determinants of nutritional intake. Third, since there are large differences in economic development levels and diets across China, the data used in previous studies covered relatively few areas [22,23,24,25,29].

Thus, we used a larger sample with more provinces to control for geographical heterogeneity. Therefore, the main aim of this study was to evaluate the impacts of internet access on nutritional intake in Chinese rural residents. Particularly, this article provides answers to the following questions: what is the difference in nutritional intake between rural residents with and without internet access? What is the potential mechanism by which the internet influences nutritional intake?

The remainder of the article is organized as follows: Section 2 provides the background of internet development in rural China. Section 3 introduces the materials and methods, and the empirical results are shown in Section 4. Section 5 further investigates the potential channels through which internet access affects nutritional intake. Section 6 discusses the implications and limitations of our empirical results. The final section presents our conclusions.

## 2. The Use of the Internet in China

The number of internet users in rural China has rapidly increased with income growth and policy support, such as the “Broadband China” strategy implemented in 2013. Figure 1 reveals the number of rural netizens in China, which increased from 156 million in 2012 to 222 million in 2018, with an annual growth rate of 6.06% [31]. However, China’s internet market is still dominated by urban areas. In 2018, the number of urban netizens increased to 607 million [31].

Given the large rural population in China, the proportion of rural netizens in the rural population was 39.4% in 2018, while the proportion of urban netizens in the urban population reached 73.0% (Figure 2) [31,37]. Although mobile ownership is widely used as a proxy for internet access in China [22,25,32], a large number of mobile phones are not connected to the internet. The NBSC showed that the number of mobile phones owned per 100 rural households was 244.3 in 2016; however, only 47.8% of these mobile phones were connected to the internet [33]. Meanwhile, the number of computers owned per 100 rural households was 32.2 in 2016 [33], and the proportion of rural netizens in the rural population was 34.1% in 2016 [38]. These statistics indicate that in rural China, computer ownership is a better proxy for internet access than mobile phone ownership.

## 3. Materials and Methods

### 3.1. Study Type

We adopted the econometric model to empirically analyze the impact of internet access on nutritional intake in Chinese rural residents. The flowchart of the study steps is shown in Figure 3. In this study, an IV-Probit-based propensity score matching method was used.

### 3.2. Study Design

Rural residents’ access to the internet is a self-selection process. It could be affected by certain unobserved attributes, including social networks and innate abilities and motivations, which may be correlated with their nutritional intake [39,40,41]. According to the Crown, these problems may cause selection bias and produce endogeneity [42]. Thus, this study adopted a PSM method, which is a semiparametric technique and is widely used to solve the problem of selection bias and endogeneity [41,43,44,45,46]. In fact, the PSM method estimates the treatment effects between the treatment group and a matched control group of observed characteristics based on propensity scores [47,48,49]. The propensity score may be defined as the conditional probability of assignment to a particular treatment given a vector of observed covariates [45]. In this study, the rural residents with internet access are the treatment group. Those without internet access are the control group. A Probit model was constructed to estimate the propensity scores. The Probit model is given as
(1)$$p\left(x_{i}\right)=\mathrm{prob}\left(Y_{i}=1|x_{i}\right)=\int_{-\infty }^{\beta_{i}x_{i}}\phi \left(t\right)dt=\Phi \left(\beta_{i}x_{i}\right)$$
where $p\left(x_{i}\right)$ is the probability that rural resident (i) has internet access, $Y_{i}=1$ indicates that rural resident *i* has internet access, and $Y_{i}=0$ indicates that resident *i* does not have internet access. The $x_{i}$ are the relevant factors that affect internet access and mainly include the individual characteristics of rural residents, such as the gender of the household heads (HHs). It also includes the characteristics of rural households, such as per capita annual income. $\beta_{i}$ are the vectors of parameters that need to be estimated.

There was potential endogeneity due to the unobservable characteristics and simultaneity bias, which is the major limitation of the PSM. Access to the internet can increase the income of rural residents [23], and conversely, the income level of rural residents can affect internet access. Therefore, we used instrumental variables (IVs), including “the location relationship between villages and towns” and “the per capita annual income of the village”, to construct an IV-Probit model to solve such problems [50,51]. The two variables were used as instruments because they correlate with the per capita annual income of rural residents and because they do not affect the internet access of rural residents. Although rural residents in the same county have similar internet access levels, the level of internet development can be similar or different between different counties. Thus, we clustered the data by county to obtain robust standard errors.

To further obtain robust matching results, this study used three common matching algorithms, i.e., the five-nearest-neighbors matching algorithm, the kernel matching algorithm, and the radius matching algorithm. By controlling the selection bias and circumventing the endogenous problem, the unbiased estimation of the ATT was obtained. The ATT for nutritional intake is given as
(2)$$\mathrm{ATT}=E\left(Y_{i}^{T}-Y_{i}^{C}|T=1\right)=E\left(Y_{i}^{T}|T=1\right)-E\left(Y_{i}^{C}|T=1\right)$$
where $Y_{i}^{T}$ and $Y_{i}^{C}$ represent the nutritional intake of the treatment and control groups, respectively.

### 3.3. Data Collection

We estimated the internet effects using the 2012–2018 Survey for Agriculture and Village Economy (SAVE) data collected by the Institute of Agricultural Economics and Development (IAED), the Chinese Academy of Agricultural Sciences (CAAS). There were no ethical issues relating to the survey or our study. After data cleaning by excluding all samples that did not report their income or nutritional intake, 10,042 samples from six provinces (i.e., Hebei, Jilin, Fujian, Shandong, Henan, and Yunnan) remained (Figure 4). The data include 1445 samples from 2012, 1820 samples from 2013, 1610 samples from 2014, 1471 samples from 2015, 1494 samples from 2016, 1127 samples from 2017, and 1075 samples from 2018. Therefore, the data used in this study were unbalanced panel data. 

On the basis of the China Food Composition, we divided the food consumed by rural residents into the following 10 categories: cereals, edible oil, red meat, poultry, eggs, aquatic products, dairy products, vegetables, fruits, and tubers [52]. They were converted into energy (kcal), protein (g), fat (g), and carbohydrate (g) based on the nutrition table. As cereals and tubers are both staple foods, they were combined into staple foods to analyze the quantities and prices of food consumption.

### 3.4. Sample Grouping

Rural households with at least one computer were considered to have internet access (treatment group), and those without a computer were the control group, i.e., without internet access. The treatment group included 3307 rural residents with internet access and 6735 rural residents without internet access.

### 3.5. Data Analysis

In this study, the nutritional intake, i.e., the intake of energy, protein, fat, and carbohydrate, were dependent variables. Internet access was the core independent variable, and the main control variables included the household characteristics (i.e., gender, age, years of education, occupation, and agricultural training), households (i.e., the proportion of children under the age of 14, the proportion of seniors over the age of 65, and per capita per annum income), and village characteristics (i.e., per capita per annum per village income and the location). Statistical analysis was performed using Stata 16.

The descriptive statistics of the sample data are given in Table 1. The average per capita daily intakes of energy, carbohydrate, protein, and fat were 2100.53 kcal, 307.43 g, 75.50 g, and 52.47 g, respectively. Meanwhile, the average per capita income was CNY 8470. The average age and years of education of the household heads were 51.43 and 7.73, respectively.

The differences in the main variables between the treatment group and the control group, according to the results of the *t*-test, are listed in the last column of Table 1. The treatment group demonstrated a significantly lower consumption of staple foods than the control group and significantly higher consumption of the other eight food categories. The results reveal that the daily energy intake was 2142.55 kcal in the treatment group, which was significantly higher than that of the control group (i.e., 2079.89 kcal). The daily intakes of protein and fat in the treatment group were 54.65 g and 79.66 g, respectively, and both were significantly higher than those in the control group (51.40 g and 73.45 g). The daily intake of carbohydrates did not differ between the two groups. Furthermore, it was found that internet access may significantly promote the intake of protein, fat, and energy.

### 3.6. Data Quality

Compared with the NBSC (Table A1), the household food consumption of staple foods, red meat, poultry, aquatic products, dairy products, vegetables, and fruits was lower than the NBSC standard, while the consumption of edible oil and eggs was higher.

## 4. Results

### 4.1. Results of the PSM

We used an IV-Probit model to estimate propensity scores, and the results are given in Table A1. First, the result of the Wald test for exogeneity contradicted the null hypothesis of no endogeneity. Second, the F statistic value of the first stage of the IV-probit model was 99.56, which was greater than 10. It indicates that the null hypothesis of weak IVs can be rejected [53]. 

The effects of internet access on nutritional intake were estimated using the PSM method based on the five-nearest-neighbors matching algorithm (Table 2). The results showed that the daily intakes of protein and fat in the treatment group were significantly higher (by 5.02% (2.61 g) and 4.33% (3.30 g), respectively) than those in the control group. Additionally, the intake of energy in the treatment group was significantly higher (by 1.35% (28.62 kcal)) than that of the control group. However, the daily intake of carbohydrates did not differ between the two groups. 

The estimated results proved that internet access increased the proportions of protein and fat intake, which were significantly higher than the proportion of energy intake. This result may be explained by the fact that the energy was mainly supplied from carbohydrate sources. The proportion of carbohydrates reached 55–65%. In contrast, for fat, the proportion only reached 20–30% [8]. Therefore, the increases in the intake of protein and fat did not cause the same level of increase in energy. Finally, internet access significantly improved the intake of the main nutritional components (i.e., protein and fat) of rural residents.

### 4.2. Balancing, Sensitivity, and Robustness Tests

To ensure that the matching estimators correctly identify the treatment effects, the matching balancing condition and the conditional independence condition must be satisfied [45]. The matching balance was tested based on three alternative algorithms. Table 3 shows no significant differences between the treatment group and the control group after matching using the five-nearest-neighbors algorithm. However, as shown in column 2 of Table 3, if the kernel algorithm with a bandwidth of 0.06 was used, there were no differences across the two groups except in regard to the gender of HHs. Furthermore, matching results using the radius algorithm failed the balancing test. Therefore, the five-nearest-neighbors matching algorithm was preferred over the other algorithms.

Although it is difficult to directly test the conditional independence condition, the Rosenbaum bounds test was used to assess the sensitivity of the PSM method to unobserved variables [46]. The results of the Rosenbaum bounds test are shown in Table A2. It was found that the matching results were not sensitive to unobserved factors, with the exception of protein. However, the IV-Probit procedure partly fixed the endogeneity problem caused by the omitted variables. There are reasons to believe that the results shown in Table 2 are reliable. 

We compared the results of different estimation techniques to test the robustness of the estimated ATTs (Table 2). The results of the robustness test showed that the signs and magnitudes of energy, protein, fat, and carbohydrate were consistent with different estimation methods, the Probit models, and the PSM algorithms. The results in Table 2 suggested that the PSM results are robust. The results estimated by the OLS method and the ordinary Probit model based on the PSM method were biased due to the problems of endogeneity being ignored (Table 3). As compared with the PSM method, the OLS method overestimated the results of protein and fat. Meanwhile, the estimated ATTs of energy, protein, and fat obtained by the Ordinary Probit model based on the PSM method were higher than those obtained by the IV-Probit based on the PSM method.

### 4.3. Test of Heterogeneity

To further analyze the heterogeneity of the matching results, this study investigated the impacts of internet access on the nutritional intake of rural residents with different incomes based on the five-nearest-neighbors matching algorithm. First, we divided the per capita annual income of rural residents into three quantiles: (1) those with an upper limit of CNY 4887.59 (low-income group); (2) those with an upper limit of CNY 12,233.90 (medium-income group); and (3) those with an upper limit of CNY 65,911.20 (high-income group). The descriptive statistics of nutritional intake and food consumption by income groups are given in Table A3.

Compared with the results of the whole sample, the impacts of internet access on the nutritional intake of those in the different income groups exhibited different features (Table 4). Specifically, in the low-income group, internet access significantly affected the intakes of energy, protein, and fat, with increases of 3.52%, 7.40%, and 10.42%, respectively. These figures are higher than those of the full sample and the other two income groups. However, there were no significant internet impacts detected regarding the carbohydrate intake in the low-income group. In the medium-income group, the intakes of protein and fat were affected by internet access, with an increase of 5.82% (higher than the full sample, but lower than the low-income group) and 3.79% (lower than the full sample and the low-income group), respectively. In contrast, energy and carbohydrate were not significantly affected. In the high-income group, only the protein intake was significantly affected, with an increase of 2.59%. However, it was lower than that of the full sample, the low-income group, and the medium-income group. 

Furthermore, the results revealed that internet access primarily affected the intakes of protein and fat in the low- and medium-income groups. The intake of energy was only affected in the low-income group. Moreover, for the high-income group, only the protein intake was affected by internet access, but the impact was less than in the other two groups. In addition, the intake of carbohydrates in the three groups was not affected by internet access, which is consistent with the full sample.

## 5. Impact Channels

The impacts of internet access on expenditure and food consumption proved to be significant [23,29]. It is suggested that expenditure and food consumption are the main channels through which internet access affects nutritional intake [54]. First, the internet can break the constraints of market access and connect closed rural areas with the market [35]. Thus, rural residents with internet access may have a stronger willingness to consume both food and other goods, even compared to those at the same income level. Second, one of the biggest advantages of online shopping is the low prices. Although the price elasticities for nutrients were negative [55], low prices may lead to an increase in nutritional intake. Unlike urban residents, rural residents were also food producers, i.e., mainly staple food producers.

### 5.1. The Channel of Expenditure

The impacts of internet access on various consumption expenditure items are shown in Table 5. Internet access significantly increased the total consumption expenditure of rural residents by 8.90% (CNY 483.18), which is in line with the conclusions of Ma et al. [23]. Specifically, internet access significantly increased consumption expenditures on food by 6.35% (CNY 188.97), suggesting that internet access affects nutritional intake by increasing the expenditure of rural residents on food. The consumption expenditures on clothing, residence, household facilities, articles, and services (HFAS); transport and communication, education, culture, and recreation (ECR); and miscellaneous goods and services (MGS) were also significantly increased with internet access, with increases of 12.90%, 16.17%, 19.68%, 23.36%, 5.03%, and 33.79%, respectively. However, rural residents with internet access spent 9.35% less on healthcare and medical services than residents without internet access. 

### 5.2. The Channel of Food Consumption

The impacts of internet access on the quantities and prices of food consumption are shown in Table 5. In terms of the quantities of food consumption, internet access had significant positive impacts on the consumption of non-staple foods, such as edible oil, red meat, eggs, aquatic products, dairy products, and fruits [6]. In contrast, internet access had negative impacts on the consumption of staple foods and no effect on poultry or vegetable consumption. 

Regarding the consumption of non-staple foods by rural residents, internet access had significant impacts on the consumption of red meat, eggs, aquatic products, dairy products, and fruits, with increases of 5.74% (1.52 kg), 21.53% (2.10 kg), 14.55% (1.01 kg), 23.35% (1.05 kg), and 14.70% (3.32 kg), respectively. However, it had little impact on edible oil consumption, only increasing its consumption by 2.93% (0.43 kg). To a certain extent, internet access increased the food access channels of rural residents by, for example, fostering the development of e-commerce in rural areas, and more food access channels help improve the dietary quality of rural residents [56,57]. 

In terms of the prices of food consumption, internet access only significantly affected the prices of poultry, aquatic products, and vegetables, which have no effect on the prices of other foods. In most of the studies on price data in China, the price data are not the actual prices of food. Rather, they reflect the unit value of food, i.e., they comprehensively reflect the quality of food, taking into account factors such as the appearance, nutrient content, flavor, and taste of food [58]. When the food consumption expenditure of rural residents increases, they may consume more high-quality food, leading to internet access having a non-significant impact on the prices of the most expensive foods.

## 6. Discussion

As obesity has gradually become an important problem in China [2,5,36,59], it is important to evaluate the impacts of internet access on the health of China’s rural residents. Despite the lack of data concerning the body mass index (BMI), we compared our results with the nutritional intakes as recommended by the *Food and Nutrition Development Outline in China* (FNDO) (2014–2020) [60] and the Dietary Pyramid as recommended by the *Dietary Guidelines for Chinese Residents* (DGCR) (2016) [61].

The FNDO (2014-2020) recommended per capita intakes of energy and protein of 2200–2300 kcal/day and 78 g/day, respectively. Since the average per capita intake of energy and protein in both the control and treatment groups were under the FNDO (2014–2020) recommendation, internet access can improve the health of China’s rural residents, especially for low-income residents.

Furthermore, as compared with the Dietary Pyramid recommended by the DGCR (2016), the structure of the nutritional intake of China’s rural residents needs to be improved. The consumption of vegetables (recommendation quantity (RQ): 300–500 g/day), fruits (RQ: 200–350 g/day), eggs (RQ: 40–50 g/day), dairy products (RQ: 300 g/day), and aquatic products (RQ: 40–75 g/day) were insufficient, while the consumption of meat and edible oil exceeded the recommended quantity (40–75 g/day and 25–30 g/day, respectively). Since internet access can significantly increase the consumption of eggs, aquatic products, dairy products, and fruits, the diet of China’s rural residents may improve with the widespread use of the internet. However, the increase in meat and edible oil consumption due to internet access may lead to potential health issues. It should be noted that the Dietary Pyramid varies from country to country and changes over time [62,63]. Thus, the impacts of internet access on health and dietary structure should be continuously reassessed.

The findings of the study have important implications for policymakers. The positive effects of internet access suggest that it is important to speed up the construction of rural telecommunications infrastructures to ensure that the majority of rural residents in China can access the internet. It is likely that the nutritional status of Chinese rural residents can be improved. Although the nutritional status of the low-income group benefits the most from internet access, reducing the cost of internet use in rural areas should be an important goal in the process of implementing the “rural vitalization strategy”.

There are several limitations to our study. First, the SAVE data only contain food consumed at home, and food consumed away from home is increasing faster than at home in China [64]. Second, because of the questionnaire design, this study only analyzed the impact of internet access on macronutritional intake, while the intake of micronutrients, such as vitamins and minerals, also play an important role in human health [8,61]. Third, we used computer ownership as the proxy for internet access, subject to the questionnaire design. Though the proportion of computer ownership is similar to the current rural internet access rates, there is still a gap between computer ownership and internet access. Thus, our study is the first attempt at shedding light on the impacts of internet access on nutritional intake in rural China. The impact of internet access on food consumed away from home and micronutrient intake should be detailed in future research.

## 7. Conclusions

In summary, we found that internet access can promote the intake of energy, protein, and fat. The widespread use of the internet in rural China can also improve the health of China’s rural residents. Our study reveals that there is heterogeneity in the intakes of energy, protein, and fat among different income groups. Furthermore, internet access plays an important role in improving the energy intake of low-income groups, which should be considered in terms of the SDG of ending malnutrition. Our findings suggest that expenditure and food consumption are the channels through which internet access affects nutritional intake. Therefore, our study confirms that the internet is an important tool with which to improve the nutritional intake and health of China’s rural residents, and the government should increase efforts to help rural residents gain access to the internet.

## Figures and Tables

**Figure 1 nutrients-13-02015-f001:**
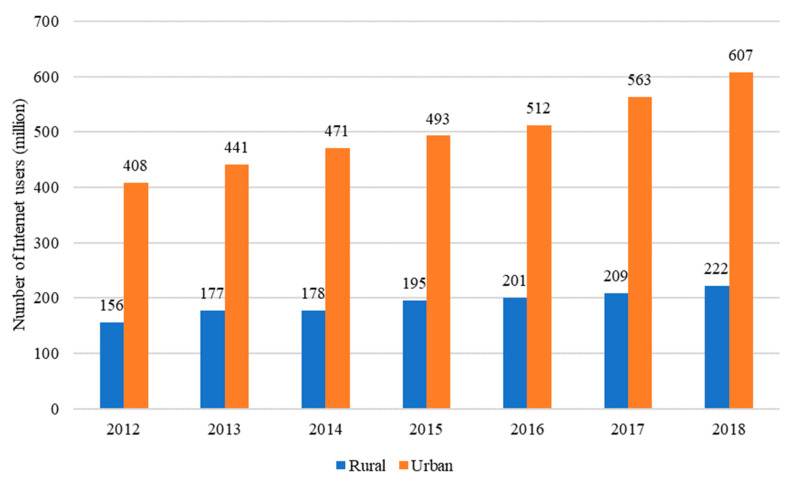
Internet users in rural and urban China. Source: The Survey Report of 2012–2014 on China’s Rural Internet Development and report numbers 34–44 of the China Statistical Report on Internet Development, issued by the CNNIC.

**Figure 2 nutrients-13-02015-f002:**
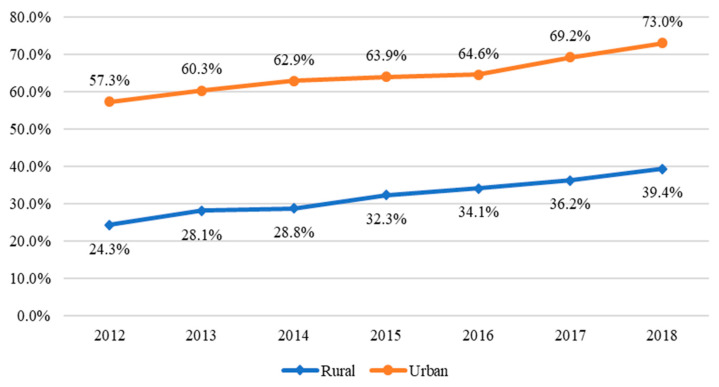
Proportion of netizens in China. Source: The Survey Report of 2012–2014 on China’s Rural Internet Development and report numbers 34–44 of the China Statistical Report on Internet Development issued by the CNNIC, and the National Bureau of Statistics of China (NBSC).

**Figure 3 nutrients-13-02015-f003:**
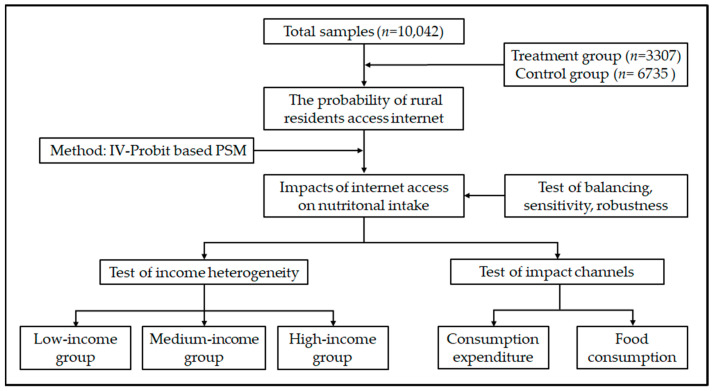
Flow chart of study steps.

**Figure 4 nutrients-13-02015-f004:**
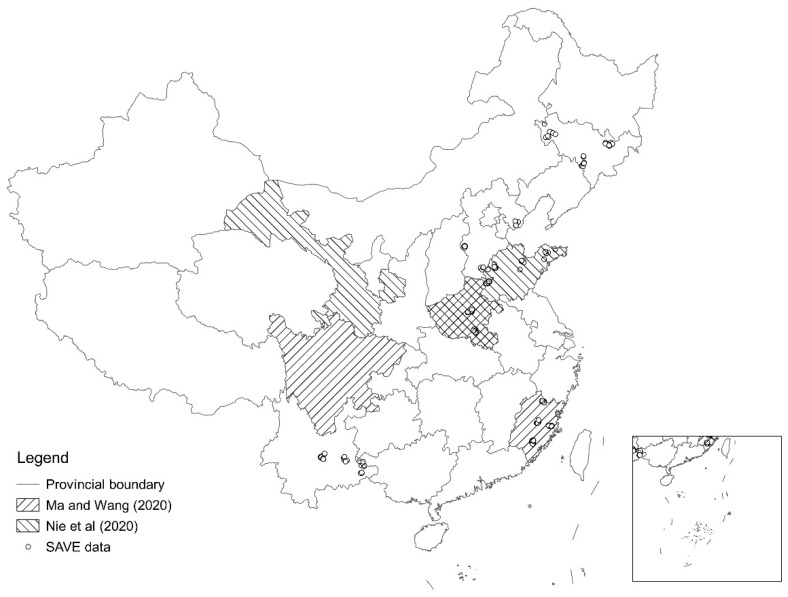
Locations of the areas selected for the field survey.

**Table 1 nutrients-13-02015-t001:** Summary statistics of basic variables.

Variables	Description	Full Sample		Treatment		Control		Diff.
		Mean	SD	Mean	SD	Mean	SD	
Household heads (HHs) characteristics								
Gender	1 = Male, 0 = Female	0.95	0.22	0.96	0.20	0.94	0.23	0.01 **
Age	Years	51.43	10.32	49.83	9.14	52.22	10.77	−2.39 ***
Years of education		7.73	2.40	8.30	2.26	7.45	2.42	0.86 ***
Occupation: only engaged in agriculture	1 = Yes; 0 = No	0.65	0.48	0.62	0.49	0.66	0.47	−0.04 ***
Agricultural training	1 = Yes; 0 = No	0.31	0.46	0.36	0.48	0.28	0.45	0.08 ***
Household characteristics								
The proportion of children under the age of 14	%	11.22	16.09	12.50	15.78	10.59	16.21	1.90 ***
The proportion of seniors over the age of 65	%	10.22	24.63	5.35	14.48	12.61	28.01	−7.26 ***
Income (per capita per annum)	CNY 1000	11.04	10.78	13.43	12.48	9.87	9.63	3.56 ***
Village characteristics								
Income (per capita per annum per village)	CNY 1000	8.47	5.51	9.79	6.14	7.83	5.04	1.96 ***
Located in the town	1 = Yes; 0 = No	0.14	0.35	0.15	0.36	0.14	0.34	0.02 *
Nutritional intake (per capita per day)								
Energy	kcal	2100.53	786.07	2142.55	806.74	2079.89	774.95	62.66 ***
Carbohydrate	g	307.43	119.06	306.29	121.57	307.99	117.81	−1.70
Fat	g	75.50	48.42	79.66	48.30	73.45	48.34	6.20 ***
Protein	g	52.47	20.74	54.65	21.80	51.40	20.12	3.24 ***
Quantities of food consumption (per capita per annum)								
Staple food	kg	138.12	51.88	135.43	51.01	139.44	52.25	−4.01 ***
Edible oil	kg	14.50	8.78	14.97	8.67	14.26	8.83	0.71 ***
Red meat	kg	25.27	24.09	27.96	25.16	23.95	23.44	4.01 ***
Poultry	kg	4.76	6.61	4.99	6.82	4.64	6.51	0.35 *
Eggs	kg	10.36	12.35	11.83	12.96	9.64	11.98	2.19 ***
Aquatic products	kg	6.74	8.94	7.91	9.64	6.16	8.51	1.75 ***
Dairy products	kg	4.80	13.20	5.52	13.13	4.44	13.22	1.08 ***
Vegetables	kg	65.02	66.08	67.45	67.91	63.83	65.14	3.62 *
Fruits	kg	23.15	25.10	25.93	28.34	21.78	23.22	4.15 ***
Number of observations		10,042		3307		6735		

Notes: *** *p* < 0.01, ** *p* < 0.05 and * *p* < 0.1; incomes were deflated with the consumer price index (CPI) provided by the NBSC (2012=100); in 2018, USD 1 = CNY 6.62.

**Table 2 nutrients-13-02015-t002:** Effects of internet access on food intake.

Daily Intake of Nutrition	PSM ^1^						PSM ^2^		OLS
	NN5 Matching ^a^		Kernel Matching ^b^		RD Matching ^c^		NN5 Matching ^a^		
	Change	Change (%)	Change	Change (%)	Change	Change (%)	Change	Change (%)	Change (%)
Energy (kcal)	28.62 *	1.35 *	32.50 **	1.54 **	61.36 ***	2.95 ***	29.47 *	1.40 *	1.91
Carbohydrate (g)	−2.90	−0.94	−1.58	−0.51	−1.77	−0.57	−3.39	−1.09	−0.73
Fat (g)	3.30 ***	4.33 ***	3.13 ***	4.09 ***	6.10 ***	8.29 ***	3.46 ***	4.55 ***	6.90 *
Protein (g)	2.61 ***	5.02 ***	2.70 ***	5.20 ***	3.22 ***	6.27 ***	2.88 ***	5.55 ***	5.77 **

Notes: *** *p* < 0.01, ** *p* < 0.05 and * *p* < 0.1; ^1^ the propensity scores calculated by the IVs-Probit model; ^2^ the propensity scores calculated by the Ordinary Probit model; ^a^ results of matching using the five-nearest-neighbors algorithm; ^b^ results of matching using the kernel algorithm; ^c^ results of matching using the radius algorithm.

**Table 3 nutrients-13-02015-t003:** The test of matching balance.

Variables	Percentage of Bias after		
	NN5 Matching ^a^	Kernel Matching ^b^	RD Matching ^c^
Gender of HHs (1 = Male, 0 = Female)	3.9	4.4 *	5.5 **
Age of HHs	−2.8	−2.3	−22.3 ***
Square of age of HHs	−3.0	−2.6	−24.5 ***
Years of education of HHs	0.1	1.6	35.0 ***
Occupation of HHs: only engaged in agriculture (1 = Yes, 0 = No)	−0.1	1.1	−8.2 ***
Agricultural training (1 = Yes, 0 = No)	−0.7	−0.1	16.5 ***
The proportion of children under the age of 14	−1.4	−2.1	11.4 ***
The proportion of seniors over the age of 65	0.3	−0.2	−29.7 ***
Per capita per annual income (CNY)	−0.4	−0.1	23.2 ***
Pseudo-R^2^	0.001	0.001	0.045

Notes: *** *p* < 0.01, ** *p* < 0.05 and * *p* < 0.1; ^a^ matching using five-nearest-neighbors algorithm; ^b^ matching using kernel algorithm with bandwidth of 0.06; ^c^ matching using radius algorithm with caliper of 0.05.

**Table 4 nutrients-13-02015-t004:** Effects of internet access on nutritional intake for different income levels.

Daily Nutritional Intake	Change (%)			
	Full Sample	Low-Income Group	Medium-Income Group	High-Income Group
Energy (kcal)	1.35 *	3.52 **	1.28	0.16
Carbohydrate (g)	−0.94	−0.43	−0.78	−1.17
Fat (g)	4.33 ***	10.42 ***	3.79 *	1.59
Protein (g)	5.02 ***	7.40 ***	5.82 ***	2.59 **
N	10,042	3348	3347	3347

Notes: *** *p* < 0.01, ** *p* < 0.05 and * *p* < 0.1.

**Table 5 nutrients-13-02015-t005:** The channels through which internet access affects nutritional intake.

Channels of Expenditure	Consumption Expenditure per Capita per Annum		Channel of Food Consumption	Quantities of Food Consumption		Prices of Food Consumption	
	Change (CNY)	Change %		Change (kg)	Change %	Change (CNY/kg)	Change %
Food	188.97 ***	6.35 ***	Staple food	−3.28 *	−2.36 *	−0.05	−1.29
Clothing	77.97 ***	12.90 ***	Edible oil	0.43 **	2.93 **	−0.12	−0.97
Residence	22.17 *	16.17 *	Red meat	1.52 ***	5.74 ***	0.12	0.46
HFAS	26.23 **	19.68 **	Poultry	0.05	1.08	−0.48 *	−2.80 *
Transport and communication	112.13 ***	23.36 ***	Eggs	2.10 ***	21.53 ***	−0.12	−1.29
ECR	33.18 *	5.03 *	Aquatic products	1.01 ***	14.55 ***	0.66 **	4.29 **
HCMS	−27.13 *	−9.35 *	Dairy products	1.05 ***	23.35 ***	0.13	1.16
MGS	49.66 ***	33.79 ***	Vegetables	2.07	3.16	0.16 **	3.64 **
Total	483.18 ***	8.90 ***	Fruits	3.32 ***	14.70 ***	0.14	1.75

Notes: *** *p* < 0.01, ** *p* < 0.05 and * *p* < 0.1; HFAS = household facilities, articles, and services; HCMS = healthcare and medical services. MGS = miscellaneous goods and services; ECR = education, culture, and recreation.

## Data Availability

The data presented in this study are available upon request from the corresponding author.

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
