# Peer review of "Internet Access and Nutritional Intake: Evidence from Rural China"

_nutrients, 2021, doi:10.3390/nu13062015_

Round 1
Reviewer 1 Report
Very innovative and relevant study which aimed to evaluate the impacts of internet access on the nutritional intake of Chinese rural residents. I have really appreciated reviewing this manuscript. It is well written and well structured, showing relevant data on the field. I have only minor suggestions to authors before it can be considered for publication:
The authors should provide some directions for future investigations at the end of the abstract.
A flowchart should be provided in section 3 with all the steps taken in carrying out the present study.
The authors should better discuss the results of the present study with the most recent and updated dietary guidelines worldwide:
Fernandez, M. L., Raheem, D., Ramos, F., Carrascosa, C., Saraiva, A., & Raposo, A. (2021). Highlights of current dietary guidelines in five continents. International Journal of Environmental Research and Public Health, 18(6), 2814.
Serra-Majem, L., Tomaino, L., Dernini, S., Berry, E. M., Lairon, D., Ngo de la Cruz, J., ... & Trichopoulou, A. (2020). Updating the mediterranean diet pyramid towards sustainability: Focus on environmental concerns. International Journal of Environmental Research and Public Health, 17(23), 8758.
The authors should provide some directions for future investigations at the end of the Conclusions.
Reviewer 2 Report
The study of Ping Xue et al., is a well designed and presented study focusing on the impact that internet access may have on food choices.
Introduction is well described and the idea is significant and authors manage to demonstrate and justify the need of the study. The obtained results actually support that internet access has an impact and the methodology followed is adecuate. However, discussion is not that well structured and there are many more results that authors do not mention or analyze.
I can observe the same for the section of conclusions. Authors need to work more on their results. This will increase the quality of the manuscript.
Reviewer 3 Report
Dear authors,
English must be reviewed and the manuscript must be adapted to the normal structure of a manuscript. The discussion should not include any results or tables results.
The study type must be included at the beginning of the methods section.
The methods must include or separate the information about "data analysis", "ethical issues".
The results are clear.
Most of the information of the conclusion (policy implications) should be moved to the discussion.
The discussion must include the limitations of the study and futures research lines.
Kind regards
Round 2
Reviewer 2 Report
Authors have done great work and have addressed the coments of the reviewers. The tables that they have added and the further explanations on discussion and results section has greatly improved the quality of the manuscript.
Reviewer 3 Report
Dear authors,
Thank you for addressing my comments.
Kind regards